

# Spectral analysis of atmospheric composition: application to surface ozone model-measurement comparisons.

Dene R. Bowdalo[1], Mathew J. Evans[1], and Eric D. Sofen[1]

[1]Wolfson Atmospheric Chemistry Laboratories, Department of Chemistry, University of York, Heslington, York, YO10 5DD, UK

*Correspondence to:* Dene Bowdalo (db876@york.ac.uk)

**Abstract.** Models of atmospheric composition play an essential role in our scientific understanding of atmospheric processes and in providing policy strategies to deal with societally relevant problems such as climate change, air quality and ecosystem degradation. The fidelity of these models needs to be assessed against observations to ensure that errors in model formulations are found and that model limitations are understood. A range of approaches are necessary for these comparisons. Here, we

apply a spectral analysis methodology for this comparison. We use the Lomb-Scargle Periodogram, a method similar to a Fourier transform, but better suited to dealing with the gapped datasets typical of observational data. We apply this methodology to long-term hourly ozone observations and the equivalent model (GEOS-Chem) output. We show that the spectrally transformed observational data shows a distinct power spectrum with regimes indicative of meteorological processes (weather, macroweather) and specific peaks observed at the daily and annual timescales together with corresponding harmonic peaks

at half, third etc. of these frequencies. Model output shows corresponding features. A comparison between the amplitude and phase of these peaks introduces a new comparison methodology between model and measurements. We focus on the amplitude and phase of diurnal and seasonal cycles and present observational/model comparisons and discuss model performance. We find large biases notably for the seasonal cycle in the mid-latitude northern hemisphere where the amplitudes are generally overestimated by up to 16 ppb, and phases are too late on the order of 1-5 months. This spectral methodology can be applied

to a range of model-measurement applications and is highly suitable for Multimodel Intercomparison Projects (MIPs).

## 1  Introduction

Ozone ($O_3$) at the surface is a pollutant, harmful to both human and plant health (Krupa and Kickert, 1989; WHO, 2005). It is the dominant source of the hydroxyl radical (OH) (Levy, 1972), which controls the concentration of key climate gases ($CH_4$, HCFCs etc.) and is an important climate gas in its own right (Forster et al., 2007).

The main sources of $O_3$ in the troposphere are from photochemical production and transport from the stratosphere. It is lost through dry deposition and photochemical loss (Monks et al., 2000; Stevenson et al., 2006; Monks et al., 2015). In the troposphere $O_3$ photochemical production is driven by the emission of precursors: nitrogen oxides ($NO_X$), carbon monoxide (CO), methane ($CH_4$) and volatile organic compounds (VOCs) which in the presence of appropriately energetic photons can





lead to a complex set of reactions which ultimately produce $O_3$ in a non-linear fashion (Ehhalt, 1999; Jenkin and Clemitshaw, 2000; Monks, 2005).

Our understanding of tropospheric ozone comes from observations of the spatial and temporal distribution of ozone and its precursors together with numerical simulations. Given the lifetime of tropospheric ozone (months) (Stevenson et al., 2006;

Young et al., 2013), global models either online (Chemistry Transport Models - CTMs) or offline (Earth System Models - ESMs) are particularly useful and are used extensively. An assessment of model fidelity is essential to find errors in processes, to evaluate where model processes are inadequate and to understand when models provide useful predictive capabilities.

Depending on the emphasis of the study, a range of methodologies have been applied to model-measurement comparisons for ozone. Many have used comparisons to 'long-term' surface ozone observations as a basis (Tanimoto et al., 2005; Jonson

et al., 2006; Oltmans et al., 2006; Derwent et al., 2008; Cooper et al., 2012; Logan et al., 2012; Hess and Zbinden, 2013; Oltmans et al., 2013; Parrish et al., 2013, 2014). Typically, these observations are averaged onto a monthly timescale and compared to a similarly averaged model output, and the two compared as a function of time. This offers some advantages. The averaged measurement and modelled datasets are small, making comparisons compact and easy to understand. It also removes the short-term variability (< monthly) that may not be of interest to the researchers.

However, this approach also suffers from a range of limitations. Processes occurring on timescales shorter than the monthly include photochemistry, deposition, transport and emission, all of which are important to the success of the model. By focusing on the monthly variability alone other timescales are ignored which may lead to insufficiently robust analysis of model performance. What is required is a methodology to assess model fidelity on a range of timescales simultaneously. Spectral methods offer this approach but for atmospheric chemistry have only been used in a limited sense, fitting standalone sine waves to time

series (Schnell et al., 2015) and applied to a small selection of coarse monthly average data (Parrish et al., 2016).

In this paper we introduce a methodology for the spectral analysis of observations of atmospheric composition data (Sect. 2). We describe this methodology for a single site (Sect. 3). We then show this methodology applied to a range of surface ozone observations sites (Sect. 4), and applied to a CTM (Sect. 5). We then compare these results and finally discuss potential reasons for biases (Sect. 6).

## 25   2   Spectral methods

The decomposition of a time-series into a set of orthogonal periodic functions was first suggested by Joseph Fourier. Classically this decomposition is into a number of sinusoidal waves each with an associated amplitude and phase. This technique is used extensively in disciplines such as engineering and geophysics. Using a computer to compute this decomposition, traditionally by correlation of basis functions with a time series, is termed the Discrete Fourier Transform (DFT). However, this method is

computationally intense which led to the development of the Fast Fourier Transform (FFT). One of the limitations of the FFT is that it is cannot accurately handle datasets with irregular time intervals. Some kind of interpolation is needed to provide data on a regular time interval which biases results (particularly at high frequencies) (Press et al., 1992; Schulz and Stattegger, 1997;





Musial et al., 2011; Rehfeld et al., 2011). Atmospheric observations inherently have irregular time intervals due to instrumental issues (power breaks, instrument failures, calibration times etc.) so another numerical method is needed.

The Lomb-Scargle Periodogram (LSP) is a spectral analysis method designed to handle gapped datasets (Lomb, 1976; Scargle, 1982; Horne and Baliunas, 1986; Press and Rybicki, 1989; Press et al., 1992). It can be formulated as a modified DFT

(Scargle, 1982; Press et al., 1992), and also equivalently by the least squares of fit of sine and cosine waveforms to a time series centred around zero (Lomb, 1976). Using the modified DFT methodology, for an equally spaced time series, taking the magnitude squared of the dot products of a time series (centred around zero) with cosine and sine waveforms at set frequencies gives a spectrum that is an estimate of the power contributing to the original data. In the presence of data gaps, the sine and cosine model functions are modified to be exactly orthogonal by an additional phase parameter $\Theta$ (Scargle, 1982), making the

estimation invariant to shifts in time of the input time series (i.e. data gaps). It is commonly represented in its normalised form (termed Power Spectral Density), e.g (Press et al., 1992), as:

$$P(\omega) = \frac{1}{2\sigma^2} \left( \frac{\left[ \sum_{i=1}^{N} y(t_i) \cos(\omega t_i - \Theta) \right]^2}{\sum_{i=1}^{N} \cos^2(\omega t_i - \Theta)} + \frac{\left[ \sum_{i=1}^{N} y(t_i) \sin(\omega t_i - \Theta) \right]^2}{\sum_{i=1}^{N} \sin^2(\omega t_i - \Theta)} \right) \tag{1}$$

where $y(t_i)$ is the observable at time $t_i$, $\omega$ is the angular frequency, and $\sigma^2$ is the variance of the time series. The phase offset $\Theta$ is calculated with the four quadrant inverse tangent, shown by Eq. (2). Additionally, the DFT is modified so when data gaps

exist the distribution of the normalised spectrum for pure Gaussian noise is exponential, equivalent to that of the equal spaced case.

$$\Theta = \frac{1}{2} \arctan \left( \sum_{i=1}^{N} \sin(2\omega t_i), \sum_{i=1}^{N} \cos(2\omega t_i) \right) \tag{2}$$

The LSP does not output any phase information natively. However, Hocke (1998) gave a method to modify the LSP algorithm to output real and imaginary components or amplitude and phase, as resultant from the Fourier transform, which we apply in

our work.

## 2.1 Spectral leakage

There are some problems in accurately identifying the amplitude and phase of periodic components. The main issue is termed 'spectral leakage'. Typically, Lomb-Scargle methods calculate power at integer frequencies equally spaced between 1 (total span of time series) and one-half of the average sampling frequency (termed 'average Nyquist frequency'), reflecting the

Fourier frequencies. If strong periodicity exists on a frequency not an integer integral on the span of the time series then its power would lie between two of the frequencies, resulting in leakage of that power throughout the rest of the spectrum. Atmospheric time series are not typically integer year long. For example, if the time series was 10.5 years long the spectrum would consist of the periods: 10.5, 5.25, 3.5, 2.1, ...,1.16 1.05, 0.955 years etc. Therefore, if large variability were contained on exactly a 1 year cycle, the LSP would spread that power throughout the spectrum.





Spectral leakage results from an assumption in spectral analysis methods that the time record is infinitely long. The transform assumes that the finite dataset is one period of an infinite periodic signal. Therefore, when the periodicity of interest is non-harmonic with the total span of the time record, there is a discontinuity, which results in leakage in the spectral domain (Horne and Baliunas, 1986).

To ensure the power leakage from multiple periodic components does not contaminate the entire spectrum, the input time series can be multiplied by a window function (Harris, 1978). The window is shaped so that it is zero at beginning and end, and has some defined shape in between. The window effectively changes the shape of the leakage in the frequency domain, limiting its impact to only a few frequencies around the peak frequency, providing a trade-off between peak resolution (the width of the peak) and spectral leakage (the amplitude of the tails of the leakage), with different windows altering the peaks

of the spectrum in different ways. In this study a Hanning window was chosen as it offers an acceptable trade-off between resolution and spectral leakage (Harris, 1978).

Although the shape of the leakage can be altered, the peak amplitude will still be underestimated still as there are still no frequencies that estimate exactly at the exact frequency of interest. However, the LSP methodology (unlike the FFT) can estimate at any frequency, allowing the exact capturing of the top of the peak. Thus, if significant cycles are known a-priori

(e.g. annual, daily etc.) their sinusoids can be calculated very accurately.

## 3   Lomb-Scargle Periodogram of surface ozone

Figure 1 shows the time series of surface ozone mixing ratio collected at Cape Verde (Carpenter et al., 2010) together with equivalent model output (see Sect. 5). Using the Lomb-Scargle methodology, this time series can be transformed into a number of sinusoidal waves at a range of periods with differing amplitudes and phases. In Fig. 2 we show the amplitude (ppb) of these

waves as a function of their period (days).

The spectrum shown in Fig. 2 has a range of characteristic features. There are broadly linear regions from 0.1 days to 10 days and from 10 days to 3000 days. There are also sets of peaks which occur at characteristic timescales (1 day, $\frac{1}{2}$ day, $\frac{1}{3}$ day etc. and 1 year, $\frac{1}{2}$ year, $\frac{1}{3}$ year etc.). We will initially discuss the identification of these linear regimes and then discuss the identification of the peaks.

### 3.1   Meteorological regimes

Figure 2 shows two distinct linear regimes in the value of the amplitude of the waves making up the LSP, which meet at around 10 days. Very similar spectra are seen in physical parameters in the atmosphere (Lovejoy and Schertzer, 2013a, b). There are 3 main scaling regimes of meteorological variability (Lovejoy and Schertzer, 2013a, b): 'weather', 'macroweather' and 'climate', with each regime being the outcome of different dynamical processes. Kolmogorov (1991a, b) suggested that

turbulent motions span a wide range of scales ranging from the macroscale at which the energy is supplied by the sun to the microscale at which energy is dissipated by viscosity. The separation between the weather and macroweather is due to the finite size of the Earth giving a physical limit to the turnover of the biggest planetary scale eddies. This is represented by the sharp





transition at around 10 days. After which there is a roughly flat frequency part of the spectrum termed 'macroweather', the power of which is the average of the largest synoptic weather systems. The 'climate' regime is only really of significance on timescales greater than 10 years and is thus not applicable to this work. Thus we end up with 2 regimes to describe the impact of meteorology on surface $O_3$ variability: weather and macroweather.

These regimes can be described by fitting a model of two joint piecewise linear functions in log-log space to the spectrum (minimising the residuals). We set the transition point at 10 days, as the theoretically maximum turnover time for the largest eddies (Lovejoy and Schertzer, 2013b). We only use periods less than 100 days, for the few points beyond this value are noisy and can often introduce significant variability into this fitting. Figure 2 shows the linear fit (green line).

To find periods which deviate from this fit, we scale this model by percentiles of the Chi-Squared probability distribution
to obtain false-alarm levels (Schulz and Mudelsee, 2002). Peaks exceeding these false-alarm levels indicate non-model components in the time series, and should be considered significant (Schulz and Mudelsee, 2002). Here, we take frequencies that have an amplitude above the 99th percent confidence level to be significant.

From Fig. 2 it is evident that there are significant peaks at the annual and half annual timescales, and at the daily, half daily and third daily timescales. For the surface ozone observational dataset described in Sect. 4 (Sofen et al., 2016) we find almost
all sites show significant peaks at the fundamentals (and most harmonics) of the annual and daily timescales. It is notable that we do not find any sites that show significance of a 7 day cycle (Altshuler et al., 1995; Marr and Harley, 2002; Beirle et al., 2003). Application of this approach to longer time series may also allow the investigation of other characteristic timescales such as NAO or ENSO (Ziemke et al., 2015).

From an atmospheric chemistry perspective these linear regimes in the spectra are interesting but are the domain of the
meteorologist. Our attention now focuses on the annual and daily peaks and their harmonics.

### 3.2   Annual and daily cycles

Figure 2 shows significant peaks at the daily and annual timescales and their harmonics. These cycles are driven by the planetary processes of the Earth's rotation around its own axis and its rotation around the sun both of which changes the predominant driving force for the atmosphere, solar radiation. Solar radiation is not strictly sinusoidal in nature, and the atmosphere is not
linear in its response. Thus the harmonics are a product of the non-sinusoidal shape of the daily and annual cycles of ozone (Valenzuela and Pontt, 2009).

#### 3.2.1   Definition of 'seasonal' and 'diurnal' cycles

For all of the sites investigated the amplitude of the daily cycle is always significantly larger than any of its harmonics. However, this is not true for the annual cycle, as the magnitude of the half annual cycle can often compete with that of the annual cycle.
Parrish et al. (2016) found the fundamental and 2nd harmonic terms solely characterise the seasonal cycle for marine boundary layer sites. To bring together the fundamental and the harmonics the we superpose the fundamental and the harmonic signals (down to the 4th harmonic - quarter cycle) to create a 'seasonal' and 'diurnal' cycle. We show an example of this in Fig. 3, where the average, 1st (fundamental), 2nd, 3rd and 4th harmonics are superposed to create the net waveform. We choose to





superpose down to the 4th harmonic as it is the highest harmonic we find significance at (i.e. the seasonal cycles in Montandon (47.3° N, 6.833° E), and Bukit Kototabang (0.2° S, 100.32° E). We characterise these cycles with an amplitude being half the peak to trough height and their phase being the timing of the maxima. We modify the LSP code to ensure we estimate at 1 and 365.25 days precisely (and 2nd, 3rd and 4th harmonics), to ensure accurate estimation of these cycles.

### 3.2.2 Fraction of total variance associated with a periodicity

The significance of the diurnal or seasonal cycles varies by location. We can calculate the fraction of the variance explained by the different waveforms (diurnal and seasonal). Taking the amplitudes and phases of the seasonal and diurnal cycles we can create periodic waveforms the span of the time series (as shown in Fig. 3). Removing the superposed waveform of the seasonal and diurnal cycles (including gaps) from the raw time series gives a time series which is solely derived of the weather and macroweather 'noise'. The variances, $\sigma^2$, of these periodic and noise time series are essentially additive so that $\sigma^2$(diurnal) + $\sigma^2$(seasonal) + $\sigma^2$(noise) = $\sigma^2$(timeseries).

We can then take the fraction of either the diurnal or seasonal cycle or both to the total variance. This ratio indicates the importance of the periodicity in explaining the total variability of the time series.

## 4 Application to observations

We apply these methods to an updated hourly version of the long-term surface ozone dataset in (Sofen et al., 2016). The data is drawn from the AirBase, CAPMON, CASTNET, EANET, EMEP, EPA AQS, NAPS, SEARCH and WMO GAW monitoring networks (see (Sofen et al., 2016) and references therein for details), and for simplicity we choose the period between 2005 and 2010 as this represents the most comprehensively observed time period. We exclude sites with data gaps of more than 365 days in this period and additionally sites with data gaps greater than 60 days in 3 or more years. We additionally limit the sites to be below 1.5km from sea-level. Figure 4 shows the location of the 710 valid sites. Most of the sites are from the US EPA AQS and EU AirBase datasets which leads to an over represent of northern continental mid-latitude locations and under represent other areas of world.

We now investigate these observations in the context of the Lomb-Scargle derived diurnal and seasonal cycles.

### 4.1 Significance of seasonal and diurnal cycle

Figure 5 shows the fraction of the variance at each site that is explained by the seasonal, diurnal and the combined waveform. For most locations the seasonal cycle represents a much larger fractional variance than the diurnal cycle.

The greatest contribution to total variance from the seasonal cycle is for the Antarctic site (85%) and the oceanic and continental Southern Hemisphere (SH) sites (40-60%). This reflects the spatial homogeneousness of these regions leading to small spatial gradients in $O_3$. Without spatial gradients to advect, weather systems cannot induce much variability, thus diurnal and seasonal variability dominates. For high $NO_X$ regions in the North Eastern US, Southern and Central Europe and Japan



(Fig. 13c), the seasonal cycle contributes 30-50% of the total variance. In Southern Central US contribution from the seasonal cycle to the total variability is very small (2-10%).

For the oceanic, polar and sites in low $NO_X$ areas in the extra-tropics (i.e. Cape Point, Cape Grim) the diurnal cycle is negligible. These diurnal cycles are typically small as ozone production / loss in these low $NO_X$ environments is small. However, it is a major contributor (30-50%) to the total variability for some low latitude regions in North America and Europe where high $NO_X$ concentrations and photolysis rates lead to significant diurnal cycles.

Superposition of the diurnal and seasonal cycles gives a measure of the fraction of total variance induced from periodicity. For most sites the percentage contribution is between 40 and 60%. The highest value being for the Antarctic site (85%). The site with the lowest % contribution from periodicity is in Indonesia (15%), almost on the equator, where there is very little variability in the solar radiation.

From this analysis it is evident that forcing of the atmosphere from seasonal and diurnal processes (changes in solar irradiation, chemistry, emissions etc.) are for responsible for the most part for around 50% of the variability seen in these sites. The remaining 50% of the variability is attributable to changes on the weather or macroweather timescales due to processes such as boundary layer mixing, synoptic systems, changing emissions etc. We now describe in more detail the seasonal and diurnal cycles seen at different locations.

## 4.2 Seasonal cycle

The seasonal cycle of ozone has been subject to much discussion (Derwent and Davies, 1994; Logan, 1985; Monks et al., 2000; Monks, 2000; Tanimoto et al., 2005; Cooper et al., 2010; Carpenter et al., 2010; Parrish et al., 2013, 2016). In general, polluted sites are suggested to show a mid-spring maxima, the exact reasons for which are still unresolved. Extra-tropical clean sites show a winter-spring maxima and tropical clean sites show a small winter maxima.

Our findings are consistent with the literature. The upper panels of Fig. 6 show the amplitude of the seasonal waveform for the observations. In general, most amplitudes are in the range of 0-15 ppb. Highly polluted sites such as the Central Valley in the US and the Po Valley in Italy show large amplitudes (up to 23 ppb). High amplitudes can also be seen in the Asian sites downwind of China, particularly to the south of Japan (up to 23 ppb).

The maxima in the observed seasonal waveforms (upper panels of Fig. 7) occurs in the spring (April, May) for most of the continental sites with a tendency for later peaks in South East Europe. The small number of continental sites in the Southern Hemisphere (SH) show peaks 3-5 months out of phase compared to the Northern Hemisphere (NH), peaking in the SH mid-winter to early spring (July - September). The SH oceanic site, American Samoa, has a winter phase (July), whereas the two NH oceanic sites have springtime phases (March and April). This is suggestive that the lower pollution associated with the SH sites generally leads to an earlier seasonal peak in $O_3$.



### 4.3 Diurnal cycle

The upper panels of Fig. 8 show the observational amplitudes of the diurnal cycle. In most of the locations this is small (0-15 ppb) with a tendency for larger amplitudes towards the tropics, where solar radiation is more intense. There are also higher amplitudes in regions with higher $NO_X$ emissions (Fig. 13c), with again the Central and Po Valleys being evident.

Significant differences between sites can be seen in the phases of the diurnal cycle (upper panels of Fig. 9). Clean sites (i.e. American Samoa, 14.300° S, 170.700° W) show a phase which peak close to dawn, reflecting photochemical ozone destruction during the day and ozone build up at night. Polluted sites (i.e. Payerne, 46.817° N, 6.933° E) show maxima in the early afternoon due to photochemical ozone production during the day.

The amplitude and phase of the diurnal and seasonal waveforms give a compact method of summarising much of the
variability seen in surface ozone sites. We now explore how a Chemistry Transport Model (CTM) simulates these observations.

### 5   Model perspective

GEOS-Chem is a global 3-D CTM driven by assimilated meteorological observations from the Goddard Earth Observing System (GEOS) of the NASA Global Modelling Assimilation Office (GMAO). The basic model is described in (Bey et al., 2001). We run version v9.01.03, using GEOS5 analysed meteorology at 2°x2.5° resolution run for 5 years between 2005 and
2010, outputting surface hourly $O_3$ in each gridbox. Global anthropogenic emissions of CO, $NO_X$, and $SO_2$ are from the global EDGAR v3.2 inventory (Olivier et al., 2005). Global anthropogenic emissions of Non-Methane VOCs (NMVOCs) are from the RETRO monthly global inventory for the year 2000, as described by Hu et al. (2015), except for ethane (Xiao et al., 2008) and global biofuel emissions (Yevich and Logan, 2003). Inventories are scaled for individual years on the basis of economic data. Regional inventories are used in certain regions where there is improved information, as described by van Donkelaar
et al. (2008). There are also inputs of $NO_X$ from additional sources i.e. aircraft (Wang et al., 1998), ships (Vinken et al., 2011) and biomass burning (Giglio et al., 2010). Inputs from lightning and soil $NO_X$ are calculated online (Yienger and Levy, 1995; Murray et al., 2012). Biogenic VOC emissions are from the global MEGAN v2.1 inventory (also calculated online) (Guenther et al., 2006). Stratospheric/tropospheric exchange is handled as a parameterised climatological representation of species sources and sinks, (McLinden et al., 2000; Murray et al., 2012).

### 25  5.1   Modelled power spectrum

The power spectrum for the modelled surface ozone at Cape Verde is shown in Fig. 2. As in the observed spectrum the weather and macroweather regimes are visibly separated at around 10 days. The model underestimates the amplitude on the shortest timescales (< 3 days). This is unsurprising given the model spatial scale (2°x2.5°, approx. 250km) and the timescale for model meteorological field updates (3 or 6 hours). The model does not attempt a detailed process level description of the boundary
layer mixing that occurs on the timescale of hours. As the timescale increases, the power in the model increases until it is comparable to that observed. This occurs at roughly 3 days. After this point the model appears to well simulate the power





spectrum for both the weather and macroweather regimes. Thus care needs to be taken in interpreting output of this model on timescales of less than around 3 days as much of the meteorological variability will be missing. In general this will be true for all models. Therefore, on some timescales the model cannot be expected to interpret the observed variability. When preparing model experiments this should be considered.

As with the observations there are peaks at 365.25 days and 1 day with appropriate harmonics. As per the observations we superpose the daily and annual fundamentals with their harmonics to produce seasonal and diurnal signals which we describe with a phase and amplitude. We now investigate the amplitude and phase of the diurnal and seasonal cycles.

### 5.2    Seasonal cycle

The lowest panel of Fig. 6 shows the modelled amplitude for the seasonal cycle in surface ozone. As with the observations, the
model shows large amplitudes over regions with significant anthropogenic $NO_X$ emissions (Fig. 13c) such as North America, Europe and Asia (up to 26 ppb). Regions with significant annual cycles in the $NO_X$ emissions, such as from biomass burning in the Amazon and Central Africa also have large cycles (up to 27 ppb). These large amplitudes can be seen to extend away from the source regions into the Pacific and Indian oceans. Over the remote tropical oceans the seasonal cycle is very small (1ppb). Due to a scarcity of observations, many of these features are unobserved.

Figure 7 shows the global seasonal phase of modelled surface $O_3$ (lower panel). There are distinct bands of phases. Over polluted NH continental regions a July-September maximum is calculated, with the cleaner northern extra-tropics showing a April-May maximum and the clean tropics a December-February maximum. In the SH there is a September-December maximum for continental regions, and a July-September maximum over the oceans and Antarctica.

### 5.3    Diurnal cycle

The largest diurnal amplitudes (lower panel of Fig. 8) are found in Eastern China (up to 29 ppb) where the emissions of $NO_X$ are greatest. This leads to large daily photochemical production of $O_3$ but also large titration by NO at night. High diurnal amplitudes are also found over the polluted North East US (13-17 ppb), Central Europe (10-13 ppb) and India (11-15 ppb). Again, regions with significant annual cycles in the $NO_X$ emissions from biomass burning also have large amplitudes i.e. Amazon, Indonesia and Central Africa.

Figure 9 shows the global diurnal phases of modelled surface $O_3$ (lower panel). As with the observations the 2 distinct clean and polluted regimes emerge. The polluted areas almost all have diurnal cycle peaking at 14:00 or 15:00. This band includes all continental regions (except Greenland and polar regions). It also includes a band across the Northern Pacific and Northern Atlantic Oceans. The clean areas almost all have a phase at 08:00, the exception being a circumpolar band of phases which peak at 04:00 around Antarctica. The diurnal phase at the poles looks incoherent, which is predominantly due to the very small
amplitudes in these regions, thus the phase becomes practically irrelevant.



## 6   Model - Measurement comparisons

The previous sections investigate the absolute amplitude and phase of the seasonal and diurnal cycle. In this section we use these parameters to investigate model performance against observations.

### 6.1   Seasonal cycle

Figure 10 shows the polar representation of the seasonal cycle for the observations, model and the difference between the two. North American and European site seasonal amplitudes are on average overestimated, (up to 16 ppb). The seasonal phase also shows biases with most sites' phases in North America and Europe peaking 1-5 months later than the observations, in mid-late summer rather than mid-late spring. Seasonal amplitudes for the African, Antarctic, Arctic, Asian, Oceania and oceanic sites are all underestimated (up to 10 ppb) but their phases show generally good agreement with the observations.

Figure 11 shows the spatial distribution of the difference for the seasonal amplitudes and phases. The biggest model over-estimations for the amplitudes (upper panels) are in regions with high very $O_3$ precursors, i.e. North East US (up to 16 ppb) and mainland Central Europe (to to 10 ppb); both generally at sites inland, away from oceanic influence. In contrast, it is the coastal and oceanic sites where the model underestimations are greatest, with the largest coming in Asia (5-10 ppb) and Eastern Canada (up to 8 ppb).

The lower panels of Fig. 11 show in mainland Europe the seasonal phases are generally 2-3 months too late in the model and 2-4 months too late in the North East/South East US. The biggest phase differences come in the Central South US with the model phases approximately 4-5 months too late (a region where the seasonal cycle contributes very little to the total variability, Fig. 5).

### 6.2   Diurnal cycle

Figure 10 also shows the polar representation of the diurnal cycle observations, model and difference. The model has some skill in determining the diurnal amplitudes. There is on average an overestimation of North American, European and Asian diurnal amplitudes (up to 17 ppb). Amplitudes for the clean oceanic sites are well estimated, with the rest of the sites in Oceania, Africa, Antarctic and Arctic displaying reasonable agreement. The model has generally good skill for simulating the diurnal phases (ignoring the polar sites), however notable biases show in the oceanic and Asian sites with the model up to 5 hours late, and up 4 hours early respectively for the groupings. Additionally for the North American and European groupings, the model simulates the vast majority of phases in a narrow band, where there is a broader grouping of phases in the observations. This may represent issues with the timing of processes such as changes to the boundary layer height which the model fails to capture, (Lin et al., 2008).

Spatially, Fig. 12 upper panels, the biggest overestimations in the amplitudes are again in regions with high emissions of $O_3$ precursors: Central Valley US (up to 17 ppb), North East US (up to 13 ppb), Japan (up to 11 ppb) and mainland Central Europe (up to 11 ppb). The biggest underestimations come in coastal regions i.e. West Coast US (up to 11 ppb) and Southern Europe (up to 10 ppb).





The lower panels of Fig. 12 show the model in the high $NO_X$ emitting regions of North East US and Central Europe to have too early a phase also (- 1-2 hours). The largest phase offsets (excluding polar sites) are found in the oceanic sites of Bermuda and American Samoa (+4 hours).

### 6.3 Possible causes of biases

A range of model biases are evident in this analysis. These may be explained by a range of model errors/uncertainties in the emissions, deposition, chemistry, photolysis rates, boundary layer mixing, stratospheric transport, tropospheric transport, resolution etc.

The most discussed uncertainties lie in the emissions. Probably the most accurate emission estimates are for North America and Europe, but even here significant uncertainties exist. Anderson et al. (2014) finds the anthropogenic US National Emissions Inventory (NEI) 2005 $NO_X$ emissions (projected to 2011) in the Mid-East US to be 51-70% too high compared with measurements taken on the DISCOVER-AQ field campaign. The NEI 2011 emissions appear to overestimated by an even larger margin. Vestreng et al. (2009) finds $\pm$8-25% uncertainties in European $NO_X$ emissions. Stein et al. (2014) also recently found wintertime systematic underestimates in NH CO by a global CTM, best offset by increases in winter CO road traffic emissions together with an improved CO dry deposition scheme.

As anthropogenic $NO_X$ decreases, the relative importance of lightning and soil $NO_X$ is much greater and the importance of low-$NO_X$ isoprene chemistry increases (Palmer, 2003; Fiore et al., 2014). Millet et al. (2008) show the MEGAN v2.1 biogenic emission inventory in the US (Guenther et al., 2006) overestimates emissions of isoprene in areas where it specifies high emission factors.

Reduced winter/early-spring photochemical removal by NO titration (Jonson et al., 2006); efficient transport of enhanced springtime $O_3$ from East Asia (Wild and Akimoto, 2001; Tanimoto, 2002; Creilson et al., 2003; Eckhardt et al., 2003; Tanimoto et al., 2005); as well as earlier peak stratospheric-tropospheric exchange to the surface, may be synergistic factors along with reduced emissions in bringing about a springtime ozone maximum for NH mid-latitude continental sites (Parrish et al., 2013).

We attempt to correlate seasonal model $O_3$ amplitude and phase biases with average 2005-2010 model $NO_X$, Fig. 13. For the seasonal cycle the greatest overestimates of the amplitude generally correlate with the highest $NO_X$ concentrations in the model (panel a), however this is not true for the largest biases in the phase (panel b). Although the phase biases are not linear with $NO_X$ emissions, from the amplitude biases it is clear that evaluation of $NO_X$ emissions would be a sensible place to start in trying to correct biases.

### 7 Conclusions

We have used a Lomb-Scargle methodology to spectrally analyse surface ozone. We find spectra with distinct relationships between amplitude and period due to meteorological processes (weather and macroweather) as well as peaks at 1 and 365.25 day timescales (and harmonics). The amplitude and phase of the periodicity associated with these timescales varies significantly between sites.





A comparison between model output and measured surface ozone spectra shows a model underestimate of the amplitudes at high frequencies due the spatial and temporal scales inherent in the model.

A comparison between of the periodic components for model and measurements shows model biases in the seasonal cycle in the mid-latitude NH, where there is a general overestimate of the seasonal amplitudes in North America and Europe of up to 16 ppb, together with delayed phase maxima by 1-5 months. We show the amplitude biases may be related to errors in the emissions of $NO_X$. We find less biases for the diurnal cycle but show the majority of amplitudes in Europe and North America to be overestimated, by up to 17ppb.

This methodology has significant scope for future use. It can be applied to a range of model-measurement applications and the associated metrics are highly suitable for Multimodel Intercomparison Projects (MIPs). We aim to apply this methodology to the ACCMIP and CCMI MIPs to explore differences in chemistry between the different CTM/ESMs.

*Acknowledgements.* We acknowledge funding from a UK NERC grant NE/R1376101. Thanks go to the data providers at the AirBase, CAPMON, CASTNET, EANET, EMEP, EPA AQS, NAPS, SEARCH and WMO GAW monitoring networks for the data to make this evaluation possible. Thanks also go to colleagues at the Wolfson Atmospheric Chemistry Laboratories, University of York for the maintaining of the monitoring station in Cape Verde, providing data for this paper.





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





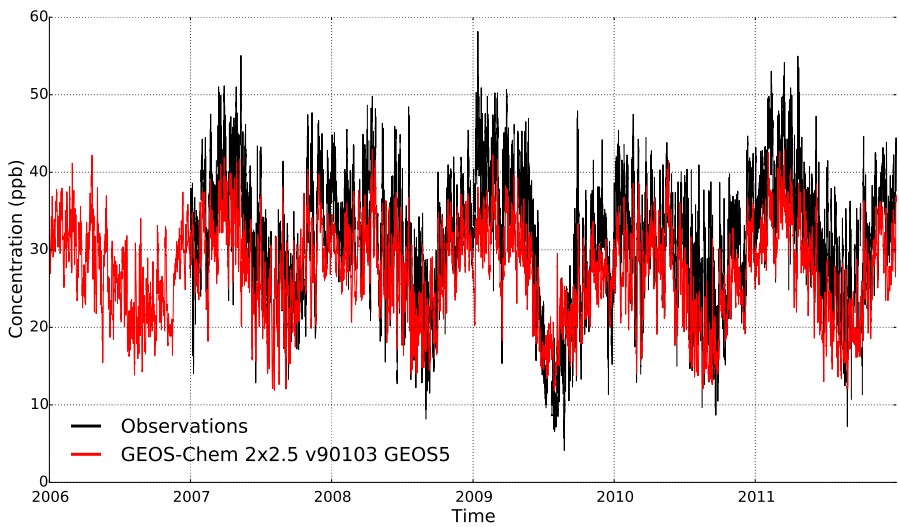

**Figure 1.** Time series of surface $O_3$ at Cape Verde for the observations (black) and the GEOS-Chem model (red).

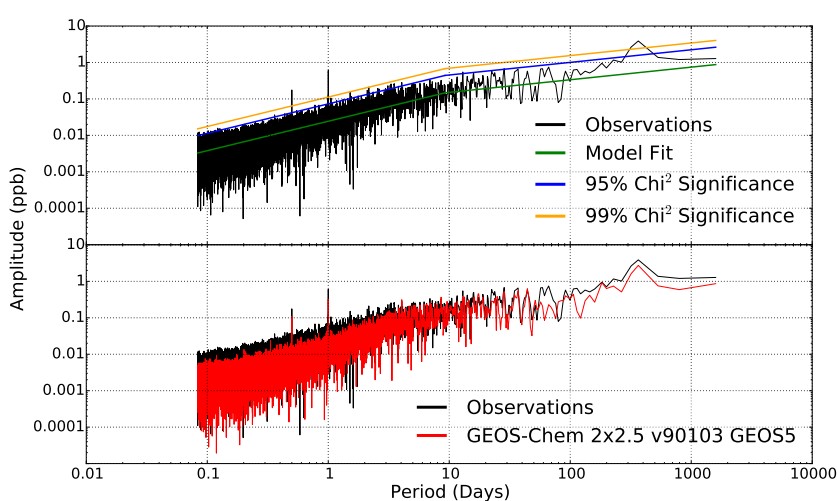

**Figure 2.** Lomb-Scargle Periodogram spectra for surface $O_3$ at Cape Verde. The upper panel shows the observed data spectrum together with chi-squared false-alarm levels for significant periodicity based on a linear piecewise fit to the spectrum. The lowest panel compares the spectra of the observations (black) and the GEOS-Chem model (red).



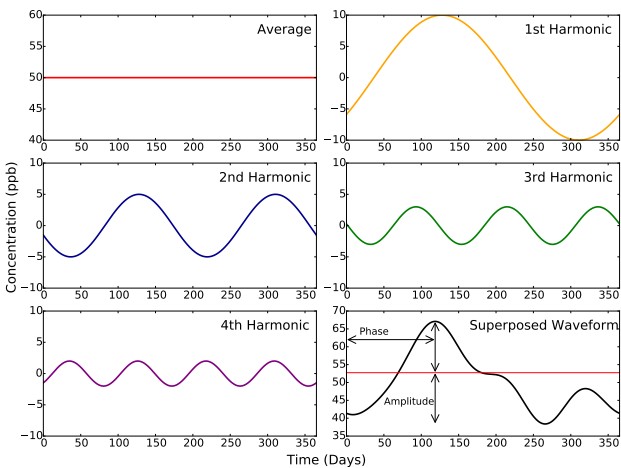

**Figure 3.** Example of spectral superposition of the average, fundamental frequency and its harmonics for a frequency of interest.

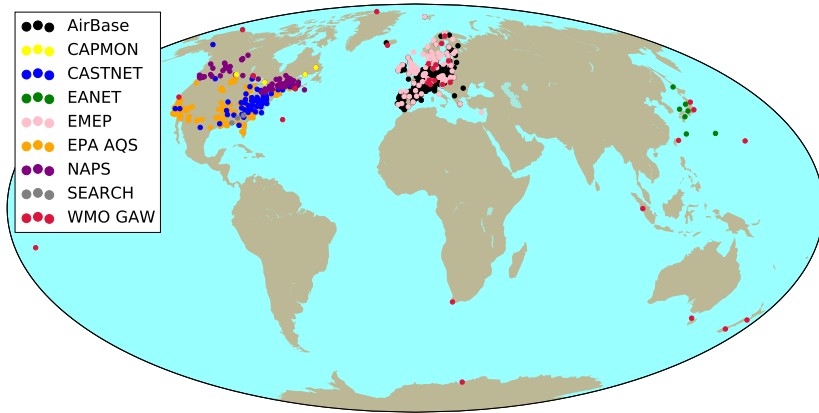

**Figure 4.** Map of surface sites reporting surface $O_3$ between 2005-2010 used in this study, coloured by the providing data network.





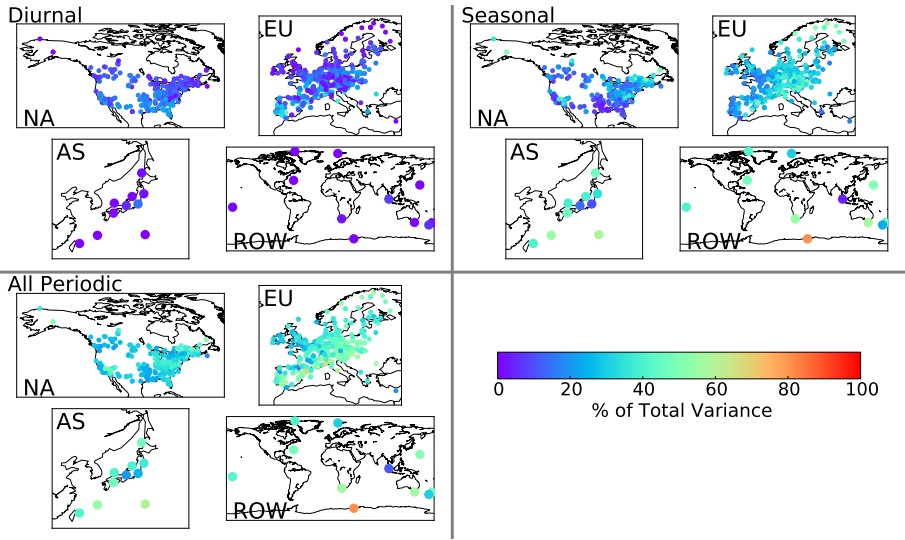

**Figure 5.** Observational fractional variance of time series by site from diurnal, seasonal and total periodicity.

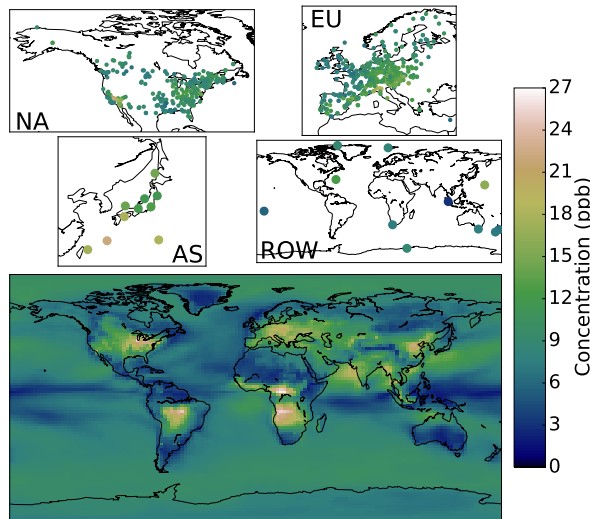

**Figure 6.** Seasonal amplitudes of observations (upper panels) and model (lower panel).





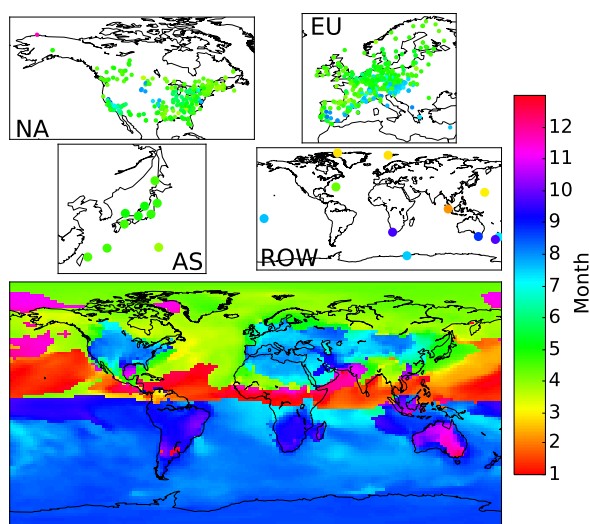

**Figure 7.** Seasonal phases of observations (upper panels) and model (lower panel).

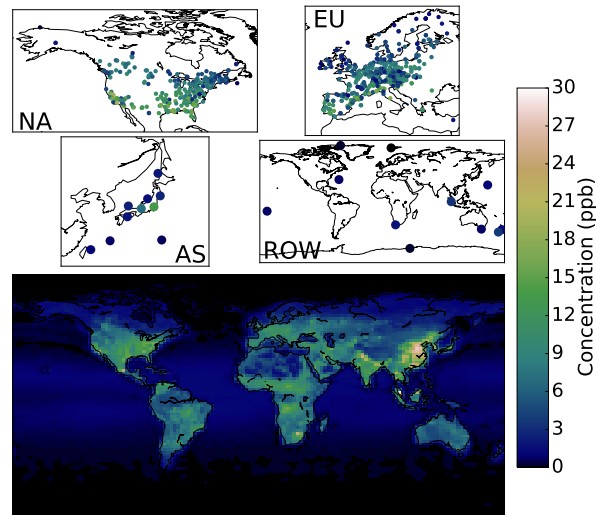

**Figure 8.** Diurnal amplitudes of observations (upper panels) and model (lower panel).





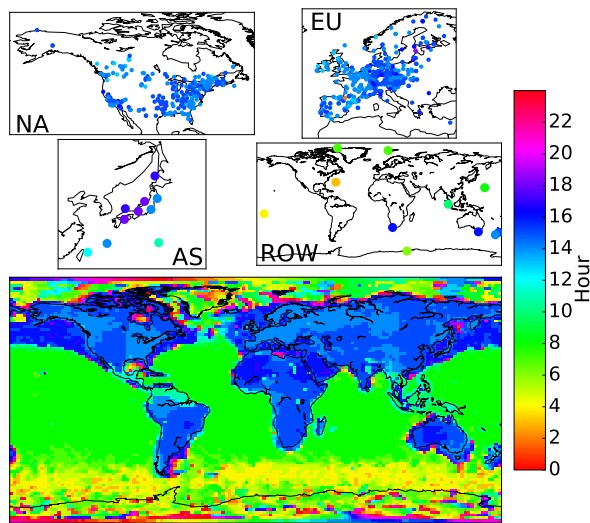

**Figure 9.** Diurnal phases of observations (upper panels) and model (lower panel).

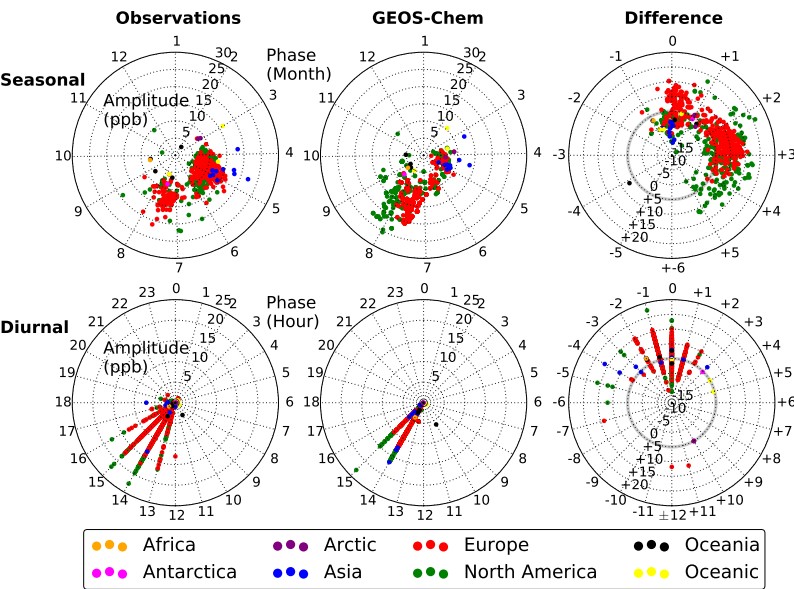

**Figure 10.** Polar plot of the diurnal and seasonal amplitudes and phases for observations and the GEOS-Chem model, and the differences between them. Circle colour indicates the location of the site.





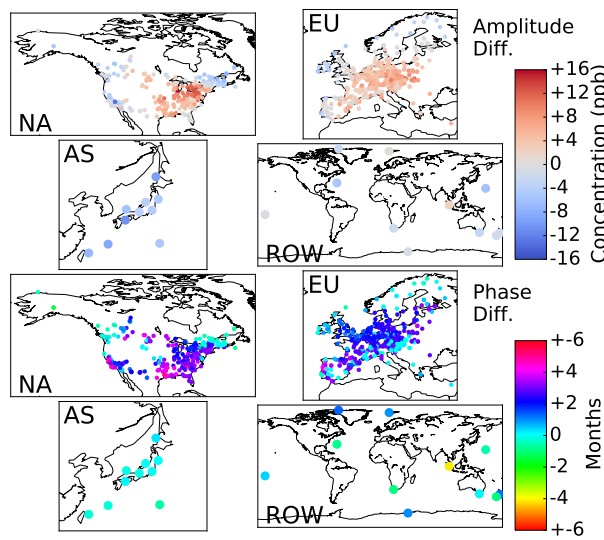

**Figure 11.** Seasonal amplitude (upper panel) and phase (lower panel) differences between observations and the GEOS-Chem model.

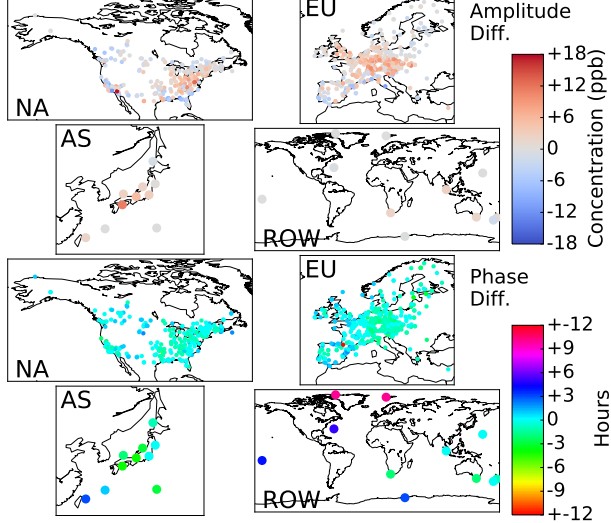

**Figure 12.** Diurnal amplitude (upper panel) and phase (lower panel) differences between observations and the GEOS-Chem model.





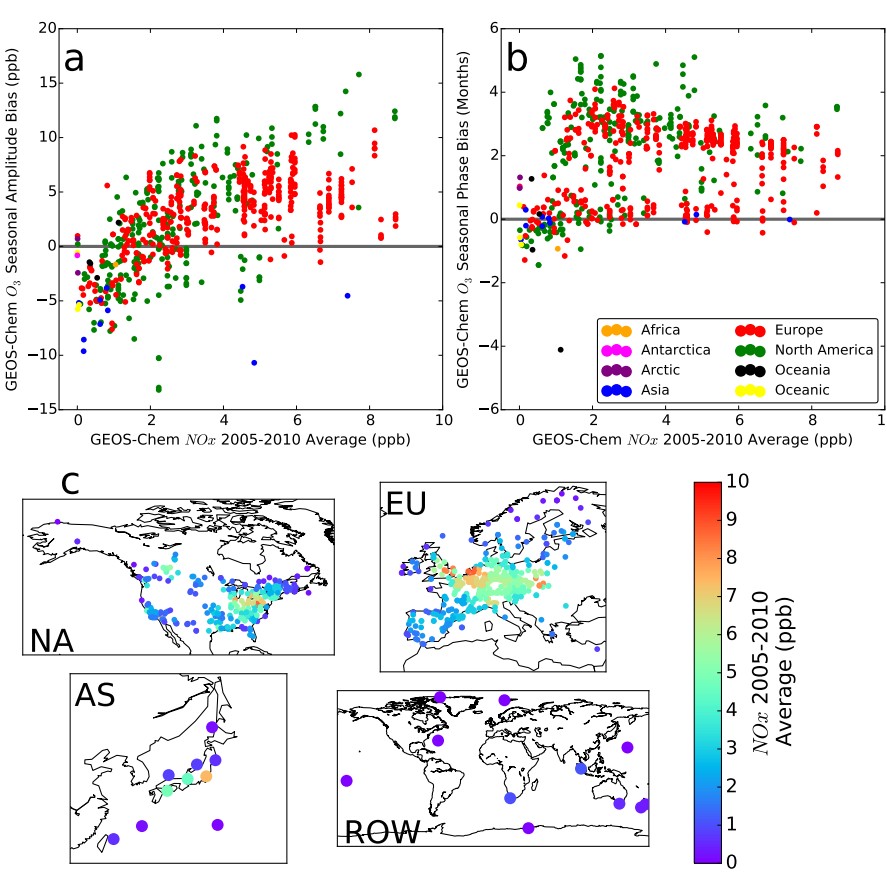

**Figure 13.** (a) Seasonal amplitude bias vs 2005-2010 Average GEOS-Chem model $NO_X$, (b) Seasonal phase bias vs 2005-2010 Average GEOS-Chem model $NO_X$, (c) 2005-2010 Average GEOS-Chem model $NO_X$ by observational site.