# Peer review of "Spectral analysis of atmospheric composition: application to surface ozone model-measurement comparisons."

_Atmospheric Chemistry and Physics, 2016_

## Referee Comment (RC1) · Anonymous Referee #1 · 12 Apr 2016

This manuscript gives a nice demonstration of the utility of spectral analysis for comparing modelled and observed surface ozone. The analysis is well done and the topic is appropriate for ACP. However before I recommend publication I would like for the authors to revise the manuscript according to my comments below.

Major comments:

You place a lot of emphasis on macroweather, a relatively new term that I wasn't even aware of. The term does not appear in the AMS Glossary (http://glossary.ametsoc.org/index.php?title=Special:AllPages/M), and I had to read about it here: http://www.cambridgeblog.org/2013/09/expect-macroweather/ . Please provide a definition of macroweather and state how it differs

from climate. Along these same lines, your description of turbulence and macroweather at the end of page 4 seem to be outside the standard definitions. Looking at the various definitions of turbulence in the AMS Glossary (http://glossary.ametsoc.org/w/index.php?title=Special:AllPages&from=Tipping-bucket+rain+gauge) turbulence seems most applicable to the microscale (e.g. it is calculated using the Reynolds number which describes viscosity). The term eddy is also most often used in reference to small scale motions. And here you seem to imply that macroweather encompasses the definitions of synoptic-scale troughs and anticyclones. But from my reading of Lovejoy's description, macroweather is more a time scale than a physical weather system. For example I would not refer to a single synoptic scale trough passing over Europe in the span of 3 days as macroweather, but I would consider a series of troughs and anticyclones passing over Europe during a month to be macroweather. Finally, where do the approximately 5-year ozone fluctuations caused by ENSO fall in the time-scale from macroweather to climate? For example, Lin et al. show that ozone at Mauna Loa and across the western USA is strongly affected by ENSO.

Title: Tropospheric ozone trends at Mauna Loa Observatory tied to decadal climate variability Author(s): Lin, Meiyun; Horowitz, Larry W.; Oltmans, Samuel J.; et al. Source: Nature Geoscience Volume: 7 Issue: 2 Pages: 136-143 Published: FEB 2014

Title: Climate variability modulates western US ozone air quality in spring via deep stratospheric intrusions Author(s): Lin, Meiyun; Fiore, Arlene M.; Horowitz, Larry W.; et al. Source: Nature Communications Volume: 6 Published: MAY 2015

Page 4 lines 21-24 Here you introduce the concept of your analysis using figure 2, but it does not show everything you are describing. For example you mention a scale up to 3000 days, but your figure doesn't even go to 2000. You also mention peaks at 1/3 day and 1/2 and 1/3 year, but I simply cannot distinguish such peaks. Either provide a figure that shows these peaks, or change the description.

[Figure]

Page 6 line 1 What physical process does the 4th harmonic represent... or the other harmonics?

Page 6 line 16 The EPA AQS database has hundreds of present-day sites for which many years of data are available. Yet this analysis uses just a subset. What were the criteria for limiting the AQS sites?

Page 7 lines 19 Here you almost dismissed one of the most important topics in global ozone analysis, the determination of the seasonal peak. You describe the timing of ozone at polluted sites as being "suggested". Knowledge of the timing is far more than a suggestion; we know exactly when the peaks occur just from looking at the data. The hard part is actually doing this type of analysis for many sites around the world. But fortunately you now have the analysis to show when this happens. The basic reasons for the timing of the peaks is understood as the peaks in the eastern US are now occurring earlier due to changes in emissions as shown by Clifton et al. 2014. There is also lengthy discussion of this topic in Cooper et al. 2014, who show that heavily polluted sites in the US, Europe, and especially China, still have summertime peaks. Figure 7 of Cooper et al 2014 shows the month during which tropospheric column ozone peaks, according to the OMI/MLS satellite product, which can be compared to your surface plot.

Clifton, O. E., A. M. Fiore, G. Correa, L. W. Horowitz, and V. Naik (2014), Twenty-first century reversal of the surface ozone seasonal cycle over the northeastern United States, Geophysical Research Letters, 41, 7343-7350, doi:10.1002/2014GL061378

Cooper, O. R., D. D. Parrish, J. Ziemke, N. V. Balashov, M. Cupeiro, I. E. Galbally, S. Gilge, L. Horowitz, N. R. Jensen, J.-F. Lamarque, V. Naik, S. J. Oltmans, J. Schwab, D. T. Shindell, A. M. Thompson, V. Thouret, Y. Wang, R. M. Zbinden (2014), Global distribution and trends of tropospheric ozone: An observation-based review, Elementa: Science of the Anthropocene, 2, 000029, doi: 10.12952/journal.elementa.000029

Minor comments: If no explanation is given for a comment, please insert the suggested

text into the appropriate place in the manuscript.

Page 1 line 17 Krupa and Kickert is a very outdated reference for the impacts of ozone on vegetation, please find something more recent and authoritative. You also need a current reference describing the impact of ozone on human health.

Page 2 line 4 You cite Stevenson and Young for an ozone lifetime of months, but they conclude that it's about 22 days. Please correct.

Page 2 line 27 . . .this decomposition yields a number. . .

Page 3 line 25, commas would really help here: If strong periodicity exists on a frequency, not an integer integral on the span of the time series, then

Page 4 line 12, too many stills: will still be underestimated as there are still no frequencies

Page 4 line 19 Here and throughout the paper, ozone is measured in units of ppbv and needs to be reported as such, not as ppb.

Page 5, line 7 What do you mean by eddy?

Page 6 line 1 as it is the highest harmonic for which we find significance.

Page 6 line 2 Here you give two stations as examples, but you tell the reader absolutely nothing about these stations, so what is the reader supposed to learn?

Page 6 line 21 which leads to an over representation of northern continental mid-latitude locations and an under representation of other areas of world.

Page 6 line 28 Homogeneity

Page 7 line 3 Many of your readers won't know where Cape Grim or Cape Point are located, so please add some description.

Page 7 line 4 . . .production and loss

Page 7 line 19 Better to say baseline sites rather than clean sites.

Page 8 line 24 Stratosphere/troposphere exchange

Page 9 line 4 Therefore, on some timescales the model cannot be expected to interpret the observed variability, and this limitation should be considered when preparing model experiments.

Page 11 line 20 Here the discussion is on East Asian emissions but papers by Creilson and Eckhardt are cited which focus instead on the North Atlantic. Unless these papers specifically address the impact of East Asia they should be deleted.

Page 12 line 6 Do you mean lower rather than less?

––––––––––––––––––––––––––––––

---

## Referee Comment (RC2) · Anonymous Referee #2 · 12 Apr 2016

Description: This discussion paper describes application of spectral analysis, specifically the Lomb-Scargle Periodogram (LSP), to illuminate temporal patterns in observed and modeled hourly surface ozone, globally. Compared to the Fourier transform, the LSP is better equipped to handle missing data and has the ability to estimate at any frequency. The authors walk through the spectrum for one site, Cape Verde, in detail and then present global patterns in both the amplitude and phase for 710 sites and GEOS-Chem predictions. Seasonal and diurnal processes are responsible for about 50 percent of variability in ozone, on average, while the remaining variability stems from weather and "macroweather" (>10 days) timescales. In general, modeled ozone overestimates the seasonal amplitude and produces a phase that is 1-5 months com-

pared to observations. The authors attribute these biases largely to uncertainties in emissions.

Relevance: Spectral analysis is a powerful method for detecting cyclical patterns in data, and the LSP seems especially apt for analysis of air quality data. The analysis encompasses both observed and predicted mixing ratios, facilitating inspection of model biases that is different from the usual time-domain perspective. Adopting this perspective on a global scale produces interesting insights into how the underlying processes vary at different sites across a spectrum of latitudes, climates, and anthropogenic influence.

Assessment: Comparison of spectral patterns in both observations and predictions is a strength of the work. The methods are exceptionally clearly explained, the figures are informative, and interpretation of the result is reasonable. Overall, the writing is well organized and very clear; the paper is interesting and a pleasure to read. Suggestions for improvement are listed below.

**Specific comments**

1. p. 3, lines 3-11: Without proper context, the authors seem to be implying that this is the first time the LSP has been applied to air quality data. It would be useful to cite other papers that have utilized the LSP to analyze air quality data, such as Dutton et al., Temporal patterns in daily measurements of inorganic and organic speciated PM2.5 in Denver, Atmos Environ. 2010; 44(7): 987–998.

2. p. 4, line 17: Specify the temporal resolution of the surface ozone data from Cape Verde, and describe the location in terms of where it is, its climate, and degree of human development. Why Cape Verde? Including a contrasting site with nearly opposite characteristics would provide a nice counterexample.

3. p. 5, lines 13-18: This paragraph seems like it would fit better in section 3.2, Annual and daily cycles, and some of it is redundant with material in that section.

4. p. 5, lines 13-14: "From Fig. 2 it is evident that there are significant peaks at the annual and half annual timescales, and at the daily, half daily, and third daily timescales." If amplitudes above the 99th percentile are defined to be significant, then I do not see "significant peaks" at the half annual ($\sim$182 days) or third daily ($\sim$0.3 days) timescale. There are certainly peaks in amplitude above the red line at periods of 365 days, 1 day, and 0.5 days in Fig. 2.

5. p. 5, line 32: The distinction between "seasonal" vs. "annual" cycle is not clear throughout the paper, and this would be a good place to distinguish between the two terms rigorously, or to state that they mean the same thing. Yashayaev and Zveryaev, Climate of the seasonal cycle in the North Pacific and the North Atlantic oceans , Journal of Climatology 2001; 21(4): 401-417 did a nice job of defining the annual cycle as the first harmonic only and the seasonal cycle as the sum of the annual, half annual, and harmonics.

6. p. 6, line 28: The large variance from the seasonal cycle at the Antarctic and continental Southern Hemisphere sites may also be due to low anthropogenic influence, in addition to spatial homogeneousness.

7. p. 9, line 11: "Regions with significant annual cycles. . ." This is an example of where the distinction between annual and seasonal cycles is unclear.

**Technical corrections**

8. p. 8, line 6: "peak" should be "peaks"

9. Fig. 5: Define the abbreviations NA, EU, AS, and ROW.

---

## Author Comment (AC1) · 7 Jun 2016

The authors thank the referees greatly for their comments on the manuscript. We have modified the paper based on all their suggestions and certainty feel this has improved the paper.

Reviewer #1:

Major comments:

"You place a lot of emphasis on macroweather, a relatively new term that I wasn't even aware of. The term does not appear in the AMS Glossary (http://glossary.ametsoc.org/index.php?title=Special:AllPages/M), and I

had to read about it here: http://www.cambridgeblog.org/2013/09/expect-macroweather/. Please provide a definition of macroweather and state how it differs from climate. Along these same lines, your description of turbulence and macroweather at the end of page 4 seem to be outside the standard definitions. Looking at the various definitions of turbulence in the AMS Glossary (http://glossary.ametsoc.org/w/index.php?title=Special:AllPages&from=Tipping-bucket+rain+gauge) turbulence seems most applicable to the microscale (e.g. it is calculated using the Reynolds number which describes viscosity). The term eddy is also most often used in reference to small scale motions. And here you seem to imply that macroweather encompasses the definitions of synoptic-scale troughs and anticyclones. But from my reading of Lovejoy's description, macroweather is more a time scale than a physical weather system. For example I would not refer to a single synoptic scale trough passing over Europe in the span of 3 days as macroweather, but I would consider a series of troughs and anticyclones passing over Europe during a month to be macroweather. Finally, where do the approximately 5-year ozone fluctuations caused by ENSO fall in the time-scale from macroweather to climate? For example, Lin et al. show that ozone at Mauna Loa and across the western USA is strongly affected by ENSO." We have revised Section 3.1 for enhanced clarity, including more detailed descriptions of the spectral weather, macroweather and climate regimes and altering terminology (i.e eddy). We base a lot of the terminology in this section on the work of Lovejoy & Schertzer, 2013b. We have also added comments about longer term oscillations such as ENSO.

"Page 4 lines 21-24 Here you introduce the concept of your analysis using figure 2, but it does not show everything you are describing. For example you mention a scale up to 3000 days, but your figure doesn't even go to 2000. You also mention peaks at 1/3 day and 1/2 and 1/3 year, but I simply cannot distinguish such peaks. Either provide a figure that shows these peaks, or change the description." We have amended the text description to more accurately reflect Figure 2.

"Page 6 line 1 What physical process does the 4th harmonic represent. . . or the other harmonics?" The harmonics do not necessarily independently represent any specific physical processes, rather they are a result of the periodicity of ozone on annual/daily timescales not being purely sinusoidal. This can be demonstrated by the FFT representation of a square wave, which results in a set of characteristic odd harmonics. It has been suggested the 2nd harmonic for marine boundary layer sites has a sole independent forcing, Parrish et al., 2016, however the harmonics do not have to have independent forcings by nature, they are a product of the mathematics.

"Page 6 line 16 The EPA AQS database has hundreds of present-day sites for which many years of data are available. Yet this analysis uses just a subset. What were the criteria for limiting the AQS sites?" The number of EPA AQS sites used is significantly reduced due to a number of stringent data quality checks (i.e identification of urban sites) that are outlined in Sofen et al., 2016. We have altered our description to highlight this.

"Page 7 lines 19 Here you almost dismissed one of the most important topics in global ozone analysis, the determination of the seasonal peak. You describe the timing of ozone at polluted sites as being "suggested". Knowledge of the timing is far more than a suggestion; we know exactly when the peaks occur just from looking at the data. The hard part is actually doing this type of analysis for many sites around the world. But fortunately you now have the analysis to show when this happens. The basic reasons for the timing of the peaks is understood as the peaks in the eastern US are now occurring earlier due to changes in emissions as shown by Clifton et al. 2014. There is also lengthy discussion of this topic in Cooper et al. 2014, who show that heavily polluted sites in the US, Europe, and especially China, still have summertime peaks. Figure 7 of Cooper et al 2014 shows the month during which tropospheric column ozone peaks, according to the OMI/MLS satellite product, which can be compared to your surface plot." We appreciate this is clumsily worded and have altered the description to include references which give detailed discussion on the seasonality of ozone.

Minor comments: "Page 1 line 17 Krupa and Kickert is a very outdated reference for the impacts of ozone on vegetation, please find something more recent and authoritative. You also need a current reference describing the impact of ozone on human health." Reference replaced.

"Page 2 line 4 You cite Stevenson and Young for an ozone lifetime of months, but they conclude that it's about 22 days. Please correct." Corrected

"Page 2 line 27 . . .this decomposition yields a number. . ." Amended

"Page 3 line 25, commas would really help here: If strong periodicity exists on a frequency, not an integer integral on the span of the time series, then" Added commas.

"Page 4 line 12, too many stills: will still be underestimated as there are still no frequencies" Amended.

"Page 4 line 19 Here and throughout the paper, ozone is measured in units of ppbv and needs to be reported as such, not as ppb." Changed all instances of ppb to ppbv.

"Page 5, line 7 What do you mean by eddy?" As referenced above, we have changed the terminology and description in Section 3.1 for enhanced clarity.

"Page 6 line 1 as it is the highest harmonic for which we find significance." Changed

"Page 6 line 2 Here you give two stations as examples, but you tell the reader absolutely nothing about these stations, so what is the reader supposed to learn?" Edited to add context.

"Page 6 line 21 which leads to an over representation of northern continental mid-latitude locations and an under representation of other areas of world." Amended

"Page 6 line 28 Homogeneity" Amended

"Page 7 line 3 Many of your readers won't know where Cape Grim or Cape Point are located, so please add some description." Added geographical description.

"Page 7 line 4 . . .production and loss" Amended

"Page 7 line 19 Better to say baseline sites rather than clean sites." Amended

"Page 8 line 24 Stratosphere/troposphere exchange" Amended

"Page 9 line 4 Therefore, on some timescales the model cannot be expected to interpret the observed variability, and this limitation should be considered when preparing model experiments." Amended

"Page 11 line 20 Here the discussion is on East Asian emissions but papers by Creilson and Eckhardt are cited which focus instead on the North Atlantic. Unless these papers specifically address the impact of East Asia they should be deleted." Removed erroneous papers.

"Page 12 line 6 Do you mean lower rather than less?" Amended

Reviewer #2:

Specific comments

"1. p. 3, lines 3-11: Without proper context, the authors seem to be implying that this is the first time the LSP has been applied to air quality data. It would be useful to cite other papers that have utilized the LSP to analyze air quality data, such as Dutton et al., Temporal patterns in daily measurements of inorganic and organic speciated PM2.5 in Denver, Atmos Environ. 2010; 44(7): 987–998." We have added references citing other papers that make use of the LSP for air quality data analysis.

"2. p. 4, line 17: Specify the temporal resolution of the surface ozone data from Cape Verde, and describe the location in terms of where it is, its climate, and degree of human development. Why Cape Verde? Including a contrasting site with nearly opposite characteristics would provide a nice counterexample." We have added detail outlining the temporal resolution of the data and the geographical characteristics of Cape Verde. Cape Verde is run and maintained by the atmospheric chemistry group

at the University of York, thus it provides a readily accessible dataset that precludes us having to deal with data privacy issues associated with the display of other datasets. As requested we have added a contrasting continental site situated in California.

"3. p. 5, lines 13-18: This paragraph seems like it would fit better in section 3.2, Annual and daily cycles, and some of it is redundant with material in that section." We have moved this paragraph to section 3.2, and reworded the section to remove redundant information.

"4. p. 5, lines 13-14: "From Fig. 2 it is evident that there are significant peaks at the annual and half annual timescales, and at the daily, half daily, and third daily timescales." If amplitudes above the 99th percentile are defined to be significant, then I do not see "significant peaks" at the half annual (âĹij182 days) or third daily (âĹij0.3 days) timescale. There are certainly peaks in amplitude above the red line at periods of 365 days, 1 day, and 0.5 days in Fig. 2" We have amended the description to more accurately reflect Figure 2.

"5. p. 5, line 32: The distinction between "seasonal" vs. "annual" cycle is not clear throughout the paper, and this would be a good place to distinguish between the two terms rigorously, or to state that they mean the same thing. Yashayaev and Zveryaev, Climate of the seasonal cycle in the North Pacific and the North Atlantic oceans, Journal of Climatology 2001; 21(4): 401-417 did a nice job of defining the annual cycle as the first harmonic only and the seasonal cycle as the sum of the annual, half annual, and harmonics." We attempted originally our work to make the same distinction as suggested in Section 3.2.1, the 'seasonal' cycle for example refers to the summation of all seasonal harmonics. We have reworded this section to attempt to make this distinction clearer.

"6. p. 6, line 28: The large variance from the seasonal cycle at the Antarctic and continental Southern Hemisphere sites may also be due to low anthropogenic influence, in addition to spatial homogeneousness." Added additional description.

"7. p. 9, line 11: "Regions with significant annual cycles. . ." This is an example of where the distinction between annual and seasonal cycles is unclear." We have attempted to make this distinction clearer in Section 3.2.1 as referenced above. Technical corrections

"8. p. 8, line 6: "peak" should be "peaks"" Amended

"9. Fig. 5: Define the abbreviations NA, EU, AS, and ROW." Added definitions

Please also note the supplement to this comment:
http://www.atmos-chem-phys-discuss.net/acp-2016-172/acp-2016-172-AC1-supplement.pdf

———————————————————

[Figure]

**Supplement:**

[revised manuscript text omitted]